# The Burden of Post-Translational Modification (PTM)—Disrupting Mutations in the Tumor Matrisome

**DOI:** 10.3390/cancers13051081

**Published:** 2021-03-03

**Authors:** Elisa Holstein, Annalena Dittmann, Anni Kääriäinen, Vilma Pesola, Jarkko Koivunen, Taina Pihlajaniemi, Alexandra Naba, Valerio Izzi

**Affiliations:** 1Faculty of Biochemistry and Molecular Medicine, University of Oulu, FI-90014 Oulu, Finland; elisa.holstein@gmx.at (E.H.); annalena.dittmann@oulu.fi (A.D.); anni.kaariainen@oulu.fi (A.K.); vilma.pesola@oulu.fi (V.P.); jarkko.koivunen@oulu.fi (J.K.); taina.pihlajaniemi@oulu.fi (T.P.); 2Department of Physiology and Biophysics, University of Illinois at Chicago, Chicago, IL 60612, USA; anaba@uic.edu; 3University of Illinois Cancer Center, Chicago, IL 60612, USA; 4Faculty of Medicine, University of Oulu, FI-90014 Oulu, Finland; 5Finnish Cancer Institute, 00130 Helsinki, Finland

**Keywords:** extracellular matrix (ECM), matrisome, mutations, post-translational modifications (PTM), pan-cancer

## Abstract

**Simple Summary:**

Mutations are the driving force of the oncogenic process, altering regulatory pathways and leading to uncontrolled cell proliferation. Understanding the occurrence and patterns of mutations is necessary to identify the sequence of events enabling tumor growth and diffusion. Yet, while much is known about mutations in proteins whose actions are exerted inside the cells, much less is known about the extracellular matrix (ECM) and ECM-associated proteins (collectively known as the “matrisome”) whose actions are exerted outside the cells. In particular, while post-translational modifications (PTMs) are critical for the functions of many proteins, both intracellular and in the matrisome, there are no studies evaluating the mutations impacting known PTM sites within matrisome proteins. Here we report on a large Pan-Cancer cohort spanning 32 tumor types and demonstrate the specificities of matrisome PTM-affecting mutations over the rest of the genome, also evidencing features and findings that might be relevant for prognostication and mechanistic understanding of the supportive role of the tumor microenvironment in the tumorigenic process.

**Abstract:**

Background: To evaluate the occurrence of mutations affecting post-translational modification (PTM) sites in matrisome genes across different tumor types, in light of their genomic and functional contexts and in comparison with the rest of the genome. Methods: This study spans 9075 tumor samples and 32 tumor types from The Cancer Genome Atlas (TCGA) Pan-Cancer cohort and identifies 151,088 non-silent mutations in the coding regions of the matrisome, of which 1811 affecting known sites of hydroxylation, phosphorylation, N- and O-glycosylation, acetylation, ubiquitylation, sumoylation and methylation PTM. Results: PTM-disruptive mutations (PTM^mut^) in the matrisome are less frequent than in the rest of the genome, seem independent of cell-of-origin patterns but show dependence on the nature of the matrisome protein affected and the background PTM types it generally harbors. Also, matrisome PTM^mut^ are often found among structural and functional protein regions and in proteins involved in homo- and heterotypic interactions, suggesting potential disruption of matrisome functions. Conclusions: Though quantitatively minoritarian in the spectrum of matrisome mutations, PTM^mut^ show distinctive features and damaging potential which might concur to deregulated structural, functional, and signaling networks in the tumor microenvironment.

## 1. Introduction

Propelled by next-generation sequencing (NGS) techniques and the work of large, interdisciplinary consortia such as The Cancer Genome Atlas (TCGA), cancer research has made a gigantic leap forward in understanding the molecular details of tumorigenesis and cancer-causing mutations, aberrations, and pathways and in elucidating the events that lead to supporting and promoting tumor growth and metastatization [1].

While most of the cancer research effort has been devoted to the study of tumor cell-intrinsic processes, recent years have seen a resurgence of focus on the tumor microenvironment (TME) [2,3,4,5,6] and, in particular, on the tumor “matrisome” [7,8]. Being a framework defining structural component of the extra cellular matrix (ECM), including collagens, proteoglycans and various glycoproteins, and ECM-associated proteins, including ECM-remodeling enzymes, proteins structurally or functionally related to ECM components, as well as cytokines, chemokine, and growth factors [9,10], the matrisome represents the majority of structural and functional moieties of the acellular part of the TME and enables systems biology-level studies on its regulation and alteration in cancer [11,12,13]. This, in turn, enables understanding how the two sides of a tumor cell’s plasma membrane “worlds” are connected, as all 10 hallmarks of cancers proposed by Weinberg and Hanahan [14] are directly controlled by signals coming from the ECM [13,15,16] and the TME reflects, at least partially, the genetic and molecular setup of a given tumor [17,18,19].

Protein post-translational modifications (PTM) encompass both reversible and irreversible modifications of proteins that follow synthesis and that are absolutely crucial for protein maturation and function [20,21,22]. For example, phosphorylation of a target protein by a kinase (and its subsequent activation or deactivation)—probably the most well-known of “message passing” among cell biologists—is a form of PTM [23]. Similarly, glycosylation—the most widespread form of PTM among eukaryotes [24,25]—is widely known for its crucial role in the functions and structures of the proteins, be it the triggering of a response in an effector cell after binding to the Fc portion of an antibody [23], the building of cellulose (a β-Linked homopolymer of glucose), one of the most abundant organic molecules on the planet [26], or the mediation of an enormous amount of biological signals at the physiological and the pathological level [26,27]. It is no surprise, then, that alterations in PTMs or enzymes mediating these modifications are linked to various diseases, ranging from neurodegenerative and skeletal diseases to cancer [28,29,30].

PTMs are all the more important for the ECM and the matrisome, conferring critical structural and functional features to the proteins they target [31,32]. For example, during or after collagen biosynthesis, proline and lysine residues are enzymatically converted to 3- and 4-hydroxyproline and 5-hydroxylysine, creating hydrogen bonds which stabilize collagen molecules and confer thermal stability to the nascent collagen fibrils as well as enable recognition by integrins and DDR receptors, at least as nucleation sites [33]. Similarly, glycosylation of collagens, fibronectins, laminins, and proteoglycans (and their eventual supramolecular assemblies) is crucial for recognition by, e.g., integrins and fundamental for cell adhesion and movement across a large span of physiological and developmental processes [34,35].

Altered PTM patterns are a hallmark of several diseases, including cancer [28,29,30]. Defective PTM is, in example, crucial in the deregulation of tumor-suppressor proteins such as p53, Rb, and PTEN, in the activation of oncogenic proteins such as the estrogen receptor α (ER-α) and the androgen receptor (AR), as well as in the transactivation of a plethora of tumorigenic pathways relying on chromatin remodeling and transcription factor activation [36,37,38]. Likewise, altered matrisome PTM participate to cancer and metastasis by affecting the regulation of stiffness within the TME, ablating or enabling ECM-receptor recognition, ectopically activating or inactivating functions within the matrisome proteins, enabling or suppressing the production of cryptic active domains (such as the matricryptins and matrikines) buried within larger ECM proteins and producing or hiding antigens and neoantigens whose role in cancer is still largely unknown [31,32,39].

In a recent pan-cancer analysis of mutations and copy number alterations (CNA), we evidenced several specificities of the matrisome with respect to the rest of the genome, such as a general higher frequency of mutations and alterations, the acquisition or the lack of mutations on a domain-specific basis and a robust association of specific mutations and alterations with patient survival [40]. Given the peculiar frequency of matrisome mutations in cancer [40,41] and the tremendous impact that PTM has on matrisome functions, and considering that no systematic analysis has yet produced a catalogue of disruptive mutations affecting experimentally validated PTM loci (PTM^mut^) in the matrisome, we have devised a Pan-Cancer approach to enumerate and investigate PTM^mut^ across 32 different tumor types and more than 9000 patients.

Our results show that, unlike other mutations, PTM^mut^ are poorly tolerated by the matrisome and account for only approx. 1800 mutations in total. Despite their low counts, these mutations frequently affect the functional protein domains involved in message passing, homo- and heterotypic interactions, and catalytic, enzymatic, and structural roles, hinting at a significant impact on the overall structure and function of the ECM meshwork they reside in and providing foundations to further, targeted studies on the specific molecular disturbances they induce, their role in tumorigenesis and pathogenesis and their potential as therapeutic targets.

## 2. Materials and Methods

### 2.1. Source Data

TCGA Pan-Cancer Atlas mutation data were sourced from the University of California, Santa Cruz UCSC Xena Browser hub (http://xenabrowser.net/, TCGA Unified Ensemble “MC3” mutation calls, Xena identifier: mc3.v0.2.8.PUBLIC.xena). The following PTM were considered in the study: phosphorylation, acetylation, hydroxylation, N- and O-glycosylation, methylation, sumoylation, and ubiquitylation. Experimentally validated PTM site coordinates were downloaded from Uniprot (http://www.uniprot.org/), with the exception of phosphorylation sites sourced from Phosphosite database (https://www.phosphosite.org/homeAction.action) and N-glycosylation sites from the N-Glycosite-Atlas (http://glycositedb.biomarkercenter.org/accounts/login/). Protein regions, domains, and functions were obtained from Uniprot. The matrisome gene list was obtained from the matrisome project website (http://matrisomeproject.mit.edu/) and the conversion of matrisome gene names to protein IDs was performed in Uniprot. Gene length was obtained from the package “goseq” in R. Protein–protein interactions were sourced from BioGrid (http://thebiogrid.org/). All repositories were accessed between March and August 2020 and further finally confirmed on 20 September 2020. Mutations in the matrisome of healthy individuals from solid tissues, blood or bone marrow samples were sourced from the NCI-GDC Data Portal (https://portal.gdc.cancer.gov/) on 25 February 2021.

### 2.2. Statistical Analysis

All analyses were performed in The R Project for Statistical Computing (R). All data presented in the manuscript refer to non-silent, somatic mutations (SNP + INDELS) from the following tumor types: adrenocortical carcinoma (ACC), urothelial bladder carcinoma (BLCA), breast invasive carcinoma (BRCA), cervical squamous cell carcinoma and endocervical adenocarcinoma (CESC), cholangiocarcinoma (CHOL), colon adenocarcinoma (COAD), lymphoid neoplasm diffuse large b-cell lymphoma (DLBC), esophageal carcinoma (ESCA), glioblastoma multiforme (GBM), head and neck squamous cell carcinoma (HNSC), kidney chromophobe (KICH), kidney renal clear cell carcinoma (KIRC), kidney renal papillary cell carcinoma (KIRP), liver hepatocellular carcinoma (LIHC), lung adenocarcinoma (LUAD), lung squamous cell carcinoma (LUSC), mesothelioma (MESO), ovarian serous cystadenocarcinoma (OV), pancreatic adenocarcinoma (PAAD), pheochromocytoma and paraganglioma (PCPG), prostate adenocarcinoma (PRAD), rectum adenocarcinoma (READ), sarcoma (SARC), skin cutaneous melanoma (SKCM), stomach adenocarcinoma (STAD), testicular germ cell tumors (TGCT), thyroid carcinoma (THCA), thymoma (THYM), uterine corpus endometrial carcinoma (UCEC), and uterine carcinosarcoma (UCS). Uveal Melanoma (UVM) results are included in the first part of the manuscript, but later dropped as the few cases made their interpretation inconsistent. Quantitative categorical differences were tested using a two-sided Chi-square test, while quantitative numerical differences were tested using a two-sided Mann–Whitney U test, unless otherwise specified. Linear correlation (fitting) was tested using Pearson correlation. In all analyses, a *p* value < 0.05 was chosen as the threshold for reporting significant results. Detailed descriptions of data manipulation, filtering, analytical methods, and results are provided with the code (see the Data Availability Statement).

## 3. Results

### 3.1. Genomic Features of Overall Mutations and PTM-Affecting Mutations (PTM^mut^) in the Tumor Matrisome and in the Rest of the Genome

Our analysis starts from approx. 2.3 million non-silent mutations in 9075 patients and 32 tumor types sourced from The Cancer Genome Atlas (TCGA) database [42,43]. Of these, the whole set of matrisome mutations is a sheer minority (151,088 out of 2,277,979; approx. 6.6%) (Figure 1A). Still, considering that the matrisome includes 1027 genes (out of 21,255; approx. 5% of the whole genome), this translates into a higher mean frequency of mutations/gene in the matrisome vs. rest of the genome (approx. 147 vs. 105) (Figure 1B), as already reported [40]. Further breakdown of total mutations by their expected effect shows, globally, no major differences between matrisome and rest of the genome irrespective of the algorithm used to predict the effect of the mutation (Figure 1C and Appendix A), again as already reported [40].

On this basis, we next focused on disruptive mutations affecting known PTM sites within proteins (PTM^mut^), including typical ECM PTMs such as acetylation, hydroxylation, methylation, N- and O-glycosylation, phosphorylation, sumoylation, and ubiquitylation [31,32,39].

PTM^mut^ are a small portion of all the mutations, totaling 42,733 (approx. 1.88% of all mutations) from 6303 patients across 32 tumor types (Figure 1D and Appendix A). Of these, only approx. 1811 are found in matrisome genes, thus setting the PTM^mut^/all mutations ratio at 1.19% for the matrisome and 1.92% for the rest of the genome. Interestingly, the average ratio of PTM^mut^ normalized by gene length is also slightly smaller in the matrisome than in the rest of the genome (1.05 × 10^−3^ vs. 1.31 × 10^−3^, respectively), in contrast with mutations not belonging to the PTM^mut^ group (non-PTM^mut^) which are conversely more frequent (5.06 × 10^−2^ vs. 4.12 × 10^−2^) [40]. At a more granular level, however, the comparison across different cancer types shows no significant deviation of the PTM^mut^/length ratio from the total mutation/length ratio in any tumor (Appendix A, respectively).

The total quantity of transitions and transversions in matrisome and rest of the genome is also similar for both PTM^mut^ and all mutations (Appendix A, respectively) even breaking down by mutation effect (Appendix A), marking no major differences in intracellular and extracellular proteomes. There is, however, a small and not statistically significant lack of transversions among PTM^mut^, which derives mostly from splice-site mutations in ECM-affiliated proteins (Appendix A).

Owing to the differences in the number and types of PTM sites that exist already at the baseline level between matrisome and non-matrisome proteins (Appendix A) and that might influence the observed distribution of PTM^mut^ (Appendix A), we have devised a novel measure—the *burden*—that allows for a more robust comparison between the matrisome and the rest of the genome. This measure, in fact, quantifies the % of known PTM sites that, in each protein and for the different types of PTMs, are affected by PTM^mut^ per cancer (local burdens) and the results are then averaged across the Pan-Cancer cohort to obtain a global burden.

Interestingly, the global burden of PTM^mut^ is equivalent for acetylation, while it is significantly higher in matrisome than in the rest of the genome for phosphorylation, N- and O-glycosylation (approx. 0.3, 1.6 and 8.5 folds, respectively, Figure 1E), and of course, for hydroxylation which has no correspondences outside the matrisome in the databases assessed to map baseline PTM sites (Figure 1E). Conversely, PTM^mut^ burden in matrisome is significantly devoid of sumoylation and ubiquitylation (approx. −0.7 and −0.5 folds, respectively, Figure 1E), in line with the global frequencies of the different PTMs in the matrisome. Reassuringly, these findings hold true even when comparing the matrisome against 100 random picks of the genome, in sets of the same size as the matrisome (Χ-square test, 0.0013 min to 0.02 max). Also, we notice that the local (tumor-specific) burdens (Appendix A), once analyzed across each matrisome family (collagens, ECM-affiliated proteins, ECM glycoproteins, ECM regulators, proteoglycans and secreted factors) [9,16], correlate negatively with the amount of PTM^mut^ for that family (Figure 1F), especially for ECM-affiliated proteins, ECM glycoproteins, proteoglycans, and secreted factors indicating that—apart from a few recurrent ones—each PTM^mut^ occurs only one or twice in the whole dataset.

### 3.2. PTM^mut^ in the Tumor Matrisome

The abundance of PTM^mut^ across matrisome categories varies sensibly (Appendix A), depending (as expected) on the prevalence of certain types of PTMs in the different categories [31,32,33,34,35]. For example, hydroxylation- and sumoylation-affecting PTM^mut^ are almost exclusively found in the core matrisome, where they mostly hit collagens and ECM glycoproteins, respectively (Figure 1G). On the other hand, acetylation- and phosphorylation-affecting PTM^mut^ are more abundant in matrisome-associated proteins (Figure 1G), in line with the regulatory functions of these proteins and the PTMs affected [9] and the presence of proteins with dual intracellular and extracellular localization (and functions) in this category.

The serial breakdown of PTM^mut^ counts by type, tumor, and matrisome category shows very similar patterns of mutation across the Pan-Cancer cohort, with skin cutaneous melanoma (SKCM), stomach adenocarcinoma (STAD), and uterine corpus endometrial carcinoma (UCEC) accounting for the majority of PTM^mut^ throughout the various matrisome categories (Figure 2 and Appendix A) and no clear cell- or tissue-of-origin effects. Local differences exist, however, depending on the matrisome category (Appendix A). In example, for collagens, lung adenocarcinoma and squamous cell carcinoma (LUAD and LUSC, respectively) show as many phosphorylation-affecting PTM^mut^ as STAD or more, and as double hydroxylation-affecting PTM^mut^. Keeping with the example of collagens, the 10 most frequent hydroxylation-affecting PTM^mut^ target largely the same genes (*COL1A1*, *COL3A1*, *COL5A1* and *COL14A1*) across the Pan-Cancer cohort with the noticeable addition of *COL12A1* in LUSC and UCEC and the lack of *COL5A1* in LUAD; conversely, phosphorylation-affecting PTM^mut^ show a distinctive cluster of mutations in collagen VI genes (*COL6A2*, *COL6A3*, *COL6A5*, *COL6A6*) in UCEC, LUAD, and LUSC, of collagen III, IV, VII, and XXVII (*COL3A1*, *COL4A2*, *COL7A1*, *COL27A1*) in SCKM and of the fibril-associated collagen XXII (*COL22A1*) in all of them (Appendix A).

The ten most frequent targets of PTM^mut^ per tumor type are shown in Figure 3. Unsurprisingly, the most frequently mutated matrisome genes across the Pan-Cancer cohort [40]—namely mucin 16 (*MUC16*) and filaggrin (*FLG*)—top the list of PTM^mut^-affected genes too. There are, however, significant differences between PTM^mut^ and overall mutations, with e.g., hornerin (*HRNR*, a paralog of *FLG*) being a top-10 PTM^mut^ gene in 14/31 tumor types and versican (*VCAN*), collagen III (*COL3A1*) and XIV (*COL14A1*) all mutated in 9/31 tumor types.

In line with overall matrisome mutations, again, we observe a scarce conservation of PTM^mut^ which, mostly, occur only once in the whole dataset. 19 PTM^mut^, however, occur at least once in three or more different cohorts, candidating to a role as potential “hotspots” (Appendix A and Appendix A). Of these, 16 (16/19, approx. 84% of total) are phosphorylation-PTM^mut^, with the exception of PTM^mut^ affecting sumoylation or ubiquitylation of fibrillin 2 (*FBN2*) at lysine position 1078 and acetylation of fibroblast growth factor 14 (*FGF14*) at lysine position 245. Also, 17/19 of these “hotspot” genes (approx. 90% of total) are present in the list only once, the only exceptions being *FBN2* (though the two mutations just discussed appear at the same position) and mucin 16 (*MUC16*) enlisted thrice with PTM^mut^ affecting phosphorylation at serines in position 70, 496, and 3694. Interestingly, none of the PTM^mut^ types enriching the matrisome (hydroxylation, N- and O-glycosylation) occur frequently, suggesting that mutations at the positions where these PTMs occur are poorly tolerated or, in the case of ubiquitylation-affecting mutations in protein with dual localization, that these mutations might alter the intracellular metabolism of the proteins involved. Analysis of SIFT and Polyphen results by PTM type, however, do only show minor differences between matrisome and non-matrisome PTM^mut^ (Appendix A), with the interesting exception of O-glycosylation sites which are less frequently affected by damaging mutations in the matrisome than in the rest of the genome.

The high ratio of mutations of unknown impact in both the algorithms prompted us to also calculate the ratio of non-silent to silent mutations (dN/dS), an indicator of negative selection pressure on mutations [44]. To this aim, per each gene, we calculated the dN/dS ratio of PTM^mut^ and of non-PTM^mut^ deriving a new measure (r^dN/dS^) that represents the fraction obtained dividing the dN/dS value of PTM^mut^ by the dN/dS value of non-PTM^mut^. Comparing the r^dN/dS^ values by type of PTM (Appendix A), it is evident that the matrisome is subject to more selection (lower r^dN/dS^ values) for PTM^mut^ than the rest of the genome, in line with our previous results. Also we notice that, as compared to rest of the genome, the r^dN/dS^ value of N-glycosylation and phosphorylation are particularly low (Appendix A), suggesting that these categories are under a stringent selection for PTM^mut^ which might partially explain the poor presence of hotspot mutations.

### 3.3. Functional Characterization of Matrisome PTM^mut^

To gain further understanding in the molecular and pathogenic effects of the PTM^mut^, we evaluated all the PTM^mut^ identified so far in the context of the protein domains they reside in, comparing the frequency of these mutations to non-PTM mutations in the same domains to obtain a domain-specific PTM^mut^ ratio (Appendix A). As the structure of ECM proteins is mostly modular, these domains mediate protein–protein interactions and are thus critical for proper protein functions [45]. Hence, we expect selection for or against PTM^mut^ to be critical at this level and, where possible, we compared the PTM^mut^ ratios of matrisome and non-matrisome genes by PTM type in the same domain. Results show that the frequency of PTM^mut^ varies significantly domain-wise in matrisome vs. rest of the genome, with 128 protein domains enriched for PTM^mut^ in the tumor matrisome (128/3373 domains from the Pfam database, approx. 3.79% of total, Appendix A). In addition, also the comparison of PTM^mut^/total mutation ratio in those 128 domains shows that PTM^mut^ are less tolerated than they are when the domain does not belong to a matrisome protein, especially in the case of O-glycosylation and acetylation (Appendix A), further suggesting that PTM^mut^ might have a strong negative impact on the function of the matrisome proteins they affect or their intracellular processing steps and eventual interactions, and that this might reflect in a lower fitness of the neoplastic clones harboring such mutations. On the other hand, the introduction of PTM^mut^ might, in fewer cases, alter protein functions and confer selective advantages to the harboring clones. Though this case is impossible to define here due to the lack of position-specific mapping of PTM functions in proteins, we suggest a few lines of evidence in support of the functional consequences of PTM^mut^, focusing on those domains whose PTM^mut^ frequency is at least twice in matrisome than in rest of the genome or vice versa.

First, there are differences in the enrichment of Kyoto Encyclopedia of Genes and Genomes (KEGG) pathways that point specifically to peroxisome proliferator-activated receptor (PPAR) signaling and drug metabolism in the matrisome domains targeted by PTM^mut^ (Appendix A), in line with the recent evidence [46,47,48,49,50,51], suggesting possible gains for the clones harboring these mutations in the matrisome.

Furthermore, mapping the matrisome PTM^mut^ locations into the regions and sub-regions annotated for function by Uniprot we show that, of 921 mutations annotated with region information (no information available for the reminder 986), 286 (286/921, 31%) fall in one such a region (Appendix A), again suggesting a functional consequence for matrisome PTM^mut^. At an even finer scale, we attempted mapping PTM^mut^ and non-PTM mutations into the sequence context they belonged to, focusing on both specific motifs mediating matrisome interactions with receptors and other proteins and, more in general, on the sequence features of the mutated area. For the specific motifs (GPP, GVD, RGD, LDV, GFPGER, and GLPGER) [52,53] we applied a stringent cutoff of max ±3 amino acids from the mutated residue, while for the overall sequence we fetched ±10 amino acids from the mutated residue. Interestingly, both PTM^mut^ and non-PTM mutations are unlikely to occur in any of the motifs scanned, with only 11 PTM^mut^ from 7 genes mapping (Appendix A) and 2 non-PTM mutations from 2 genes mapping. Though extremely low (speaking again in favor of poor conservation of matrisome mutations disrupting functional loci), these numbers are still robustly different (Χ-square test < 2.2 × 10^−16^), suggesting that these few PTM^mut^ are tolerated and might favor the clones harboring them. In addition, the larger sequence context also likely influences the selection in favor or against PTM^mut^, for example requiring almost exclusively a polar or hydrophobic amino acid two residues after the mutation site while showing no signs of selection for the same position in the same genes in the case of non-PTM mutations (Appendix A). As the N-glycosylation process shows evidence of stereotactic preferences for amino acids’ charge and polarity in cancer [54], it is likely that matrisome proteins—undergoing N-glycosylation more massively than the rest of the proteome—might be more subject to protein-level selection, though this cannot be determined on the sole basis of these data.

Finally, of the 437 genes harboring PTM^mut^ in protein domains whose PTM^mut^ frequency is at least twice in matrisome than in the rest of the genome or vice versa (Appendix A), 230 (230/437, approx. 53%) are involved in reciprocal homo- and/or heterotypic interactions according to BioGRID (Appendix A), further supporting functional consequences for PTM^mut^ once acquired. In this context, we notice that 39 (39/230, approx. 17% of total) of the interacting genes affected by PTM^mut^ are also among those whose mutations map into known functional regions (Appendix A and Figure 4), again sustaining the hypothesis that, once gained, these mutations might disturb the integrity of the matrisome interaction network. Our results also shed a new light on the set of mutations affecting these genes, as the evidence provided by the assessment of their mutational effect is almost exclusively “Missense mutations” and that by SIFT and PolyPhen is otherwise largely of “unknown” type. In this context, we suggest that PTM^mut^, especially in “hub” proteins [55] within the larger matrisome network, might be an overlooked class of mutations whose real effect is on the structure and stability of the network they participate to rather than the protein *per se*, though further studies will be needed to test this hypothesis.

## 4. Discussion

In recent years, the development of “omics” techniques has opened the field of ECM and matrisome to systematic cancer-specific and Pan-Cancer investigations, which have shone a new light on the roles played by the matrisome in the TME and its importance in oncogenic and pathogenic mechanisms [3,8,9,12,13,16,40,56,57,58,59,60,61,62].

While much is known about the relative expression of matrisome genes and proteins in cancer, sensibly less is known about the mutations in the tumor matrisome. In particular, apart from a few focused studies [36,63,64,65], our recent effort at characterizing matrisome mutations Pan-Cancer [40] remains the only systematic example. In this study, we focused on a class of mutations, those affecting PTM, which might potentially have a great impact on the functional and structural roles of the tumor matrisome due to the paramount importance that PTM already have in the physiological and homeostatic functions of the matrisome.

It should be noted here that TCGA data derive from bulk but sufficiently pure tumor samples [66], ruling out a confounder role for the stromal and immune content of the samples in confidently mapping the mutations to the neoplastic cells within [40].

On this basis, we have analyzed 9075 patients and 32 tumor types from TCGA Pan-Cancer cohort and identified 151,088 non-silent mutations in the coding regions of the matrisome, of which 1811 affecting known sites of hydroxylation, phosphorylation, N- and O-glycosylation, acetylation, ubiquitylation, sumoylation, and methylation PTM (PTM^mut^). As already discussed in the text, the matrisome seems less prone to accumulate PTM^mut^ than the rest of the genome according to both the frequency of PTM^mut^ per gene length and the r^dN/dS^, suggesting a higher selective pressure on these mutations and hinting at a poor representation of the matrisome—and, in particular, of its PTM^mut^—among the germline mutations that precede and predispose to cancer and rather suggesting their appearance along somatic cancer evolution [67,68], though this aspect will require further investigations into different data sets than TCGA.

This point is of particular interest, since we recently demonstrated that the matrisome accumulates more point mutations and copy number alterations (CNAs) than the rest of the genome in general [40] and thus the limited number of PTM-affecting mutations and their poor preservation is surprising, hinting at negative selection processes against this type of mutations in the matrisome. Unfortunately, a recent retrospective analysis on the timing of mutation insurgence in TCGA [69] does not help in clarifying this point, as it covers chromosome-level events rather than the intragenic mutations we are after. On the other hand, approximate time-series of colon and lung cancer show that mutational events that affect structural matrisome components such as fibronectin 1 and collagen 1 (*FN1* and *COL1A1*) as well as proteinases (*MMP2* and *ADAM10*) and other functional moieties (*NTN4*, *PCSK6*, and *SVEP1*) are likely tumor driver [70]—though, again, these data are on a different scale than that of this manuscript, so comparison is only approximate and conceptual; it is worth noticing, however, that we found PTM^mut^ for all these genes but *NTN4*, that *FN1*, *COL1A1*, *MMP2*, and *ADAM10* all harbor PTM^mut^ in functional domains and interact reciprocally and that, in the network of interactions between PTM^mut^-affected matrisome elements, these genes (especially *FN1* and *MMP2*) are major hubs, all these evidence suggesting that matrisome PTM^mut^ might significantly contribute to altered TME dynamics. Further along this line, we notice that 143 PTM^mut^ are not found at all in healthy samples from TCGA irrespective of tumor type (Appendix A) and that 9 PTM^mut^ affect “landmark” matrisome genes that deeply characterize the given tumors [71], possibly candidating the carrier genes to a more prominent role in cancer.

In this context, based on the high variability of PTMmut (which often occur only once or twice across the whole Pan-Cancer cohort) and by similarity with other matrisome mutations in general [40,63,72], we deem unlikely that any of these mutations might have a role as driver and they seem more probably the result of passenger mutations with lower historical selection [73]. More likely, the lower amount of PTM^mut^ in the matrisome is the result of the convergent structural and functional damages wrought to the proteins by both the altered PTM profile *and* the changes at the amino acid sequence level where the PTM should be [21,74,75,76]. It must be taken into consideration, in fact, that a significant portion of all PTM^mut^ we assessed (20% to 30% of total, according to either SIFT or PolyPhen) can be classified as deleterious to the protein function, involving various types of out-of-frame mutations. Additionally, the majority of PTM^mut^ affect lysine, asparagine, proline, serine, and threonine (see Appendix A), all of which are under different selective pressures [77,78,79] as targets of mutations because of their physico-chemical properties.

Still, the majority of PTM^mut^ that we have investigated has no clear effect on the protein structure, seemingly being tolerable or outside of the threshold for potentially damaging effect. We speculate that the effect of these mutations may be at the system-level scale, where the matrisome proteins harboring the PTM^mut^ might interact incorrectly (*how incorrectly* remains to be determined at the single protein level) with its partners, impairing the network structure and stability of the matrisome in the TME and potentially affecting the signaling pathways that depend or impinge on it. In this context, we have identified at least 286 PTM^mut^ falling in a functional protein region, where they might alter the activity of the protein by proximity with catalytic residues and binding sites [80]. In example, 124 PTM^mut^ in collagens (124 out of 156, 79.5%) occur within the triple helical region, where alterations in the PTM profile can easily damage the conformation and thermal stability of the proteins [81]. Similarly, the PTM^mut^ found in versican (*VCAN*) span both the alpha and beta glucosaminoglycan (GAG) attachment domains and thus impact on the addition of GAGs to the core versican protein, potentially disrupting a high number of local interactions [82], the PTM^mut^ in fibroblast growth factor 2 and fibronectin (*FGF2* and *FN1*, respectively) affect the heparin-binding domains of these proteins, necessary for cell–matrix and matrix–matrix interactions [83,84], and those in matrix metalloproteinase 2 (*MMP2*, type IV collagenase) localize in the collagen-binding domain, where they likely play a role in the regulation of enzyme activity [85]. Even more interestingly, 6 patients (Appendix A) also showed concomitant PTM^mut^ in proteins involved in heterodimeric interactions, representing a particular case of study to identify network-scale disarrays to the matrisome which, once more, are significantly less frequent in the Pan-Cancer cohort vs. co-occurring non-matrisome PTM^mut^ (Χ-square test < 2.2 × 10^−16^).

In conclusion, we believe that our results demonstrate, at the genomic level, the potential impact of PTM^mut^ on the tumor matrisome and mark a starting point to their functional characterization, enabling a more comprehensive and integrated view of this critical piece of the TME puzzle whose fine understanding may lead to significant translational applications to improve cancer patient treatment and eventually the outcome.

## Figures and Tables

**Figure 1 cancers-13-01081-f001:**
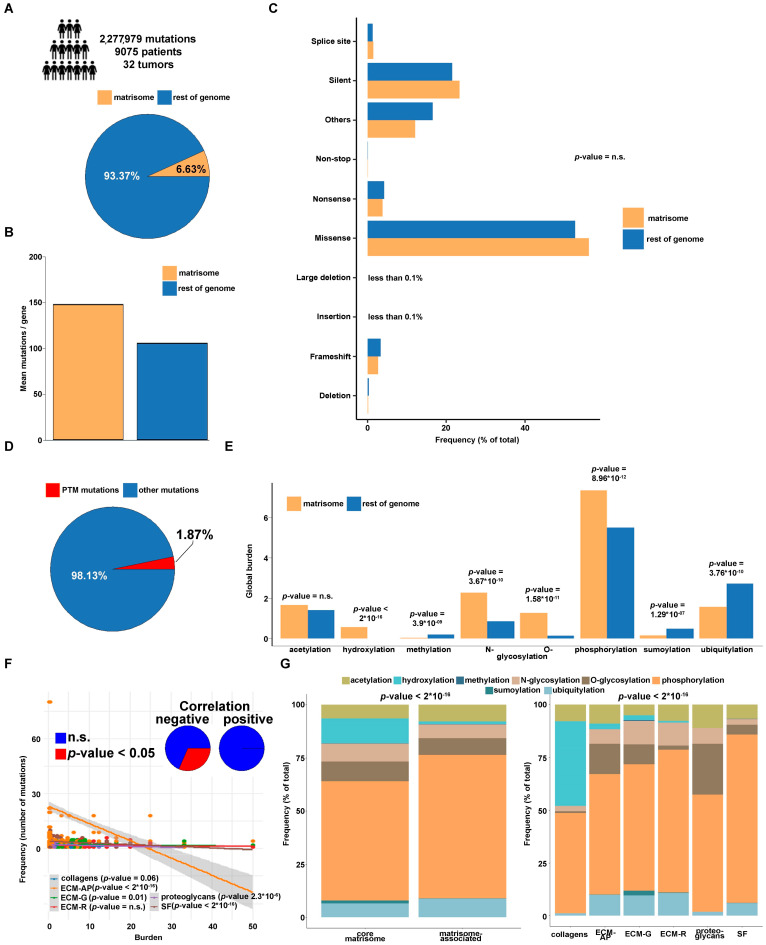
The genomic landscape of PTM-disrupting mutations (PTM^mut^) in the tumor matrisome and in the rest of the genome. Dimensions of the analysis (**A**) and comparison between the tumor matrisome and rest of the tumor genome for overall mutation frequency (**B**) and type of mutation (**C**). PTM^mut^ are a minority of all matrisome mutations (**D**) with significant enrichments or depletions of PTM types affected (**E**). The amount of PTM^mut^ across tumors and divided by matrisome family (**F**) is negatively correlated with preservation of the PTM^mut^ (indicating poor preservation of a given PTM^mut^ in multiple patients (or cancer types) and (**G**) varies considerably across the different types of PTM investigated. Abbreviations: n.s., not significant; ECM-AP, ECM-affiliated proteins; ECM-G, ECM glycoproteins; ECM-R, ECM regulators; SF, secreted factors. *p*-values are from Chi-square tests.

**Figure 2 cancers-13-01081-f002:**
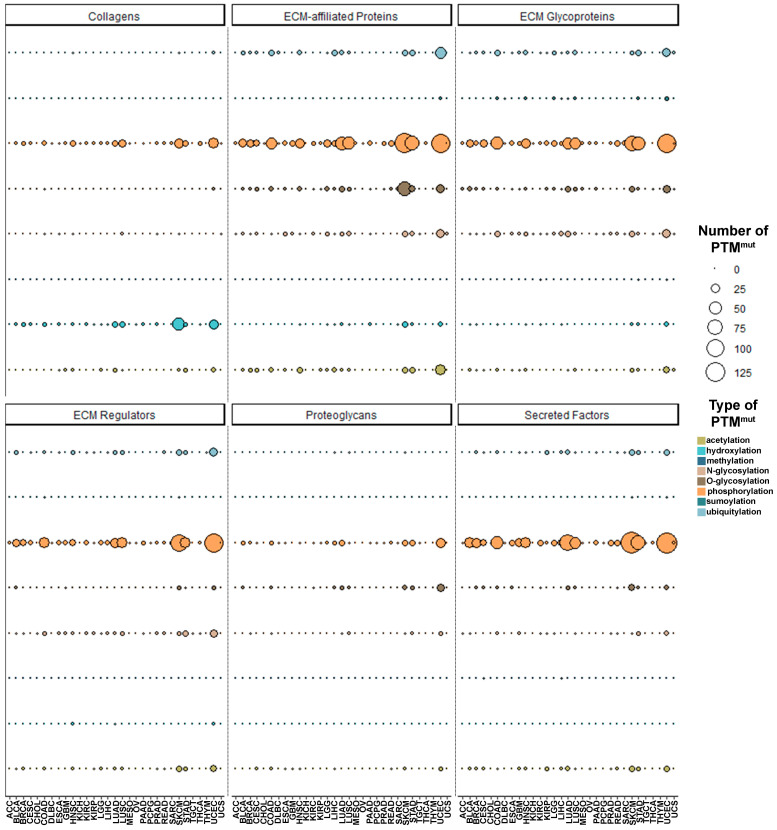
Pan-Cancer abundance and type of PTM^mut^ across the different matrisome families. The abundance (total number) of PTM^mut^, divided by the different families composing the matrisome and the different types of PTM investigated, is shown.

**Figure 3 cancers-13-01081-f003:**
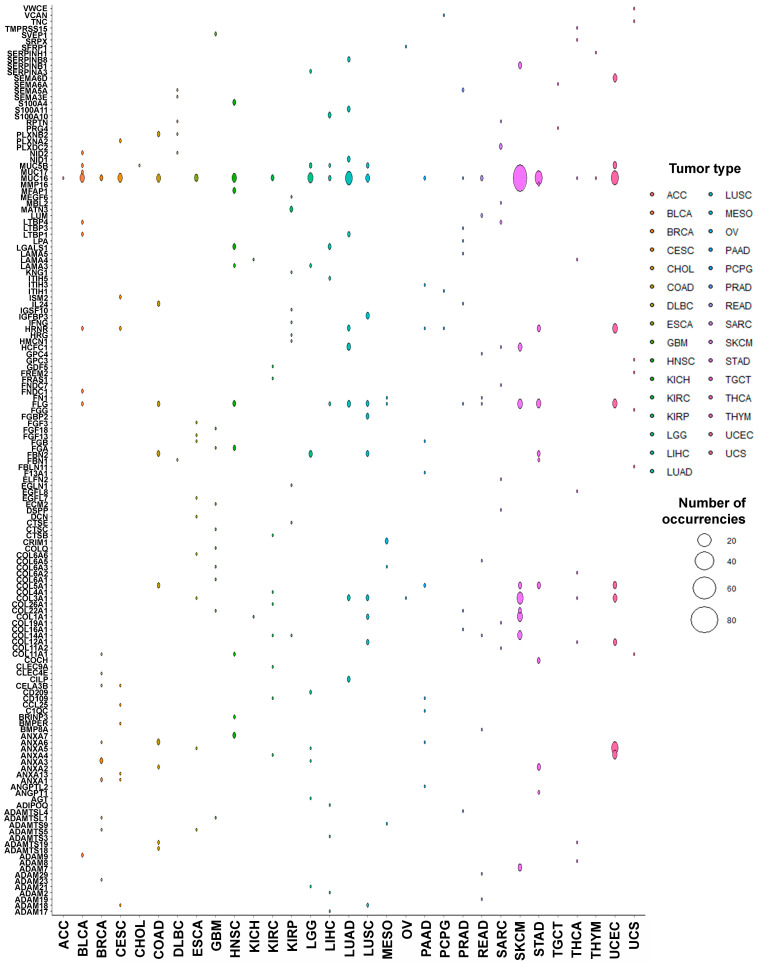
Genes most frequently affected by PTM^mut^. The abundance (total number) of PTM^mut^ for the ten most-frequently affected genes, across the different tumor types, is shown.

**Figure 4 cancers-13-01081-f004:**
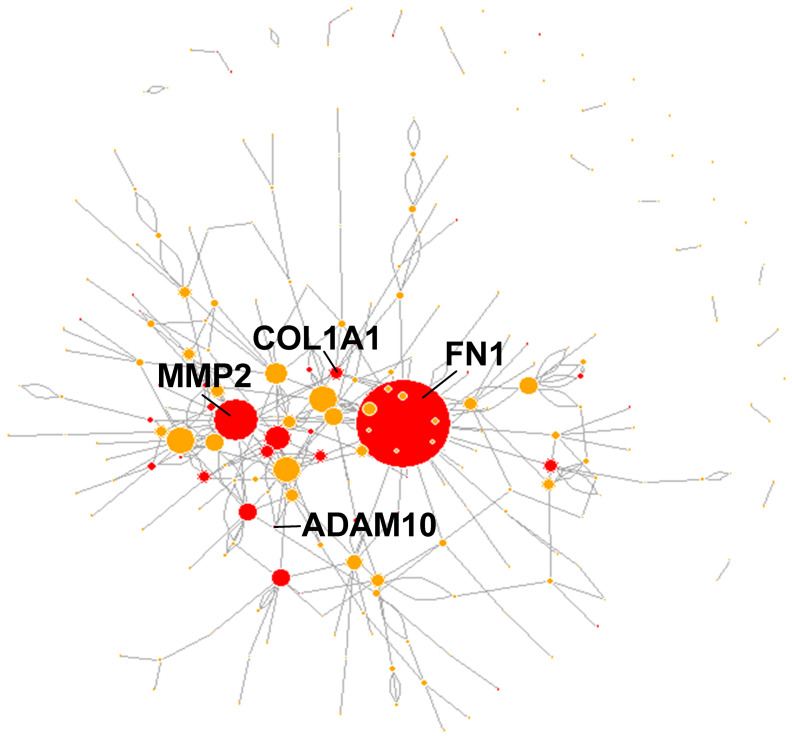
The interaction network of matrisome proteins with functional PTM^mut^. Matrisome genes that harbored PTM^mut^ and whose proteins interacted reciprocally (according to BIOGRID) were isolated and the interaction network was drawn with node size proportional to the total degree of each node. In red, matrisome genes whose PTM^mut^ mapped in in regions endowed with specific protein functions (according to UniProt).

## Data Availability

The R code sustaining this submission is available at https://github.com/Izzilab and https://rpubs.com/Izzilab/. The necessary data for the code are stored in a freely-accessible Zenodo repository (https://doi.org/10.5281/zenodo.4490484). All data were deposited on 2 February 2021.

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
