# Peer review of "The Burden of Post-Translational Modification (PTM)—Disrupting Mutations in the Tumor Matrisome"

_cancers, 2021, doi:10.3390/cancers13051081_

Round 1

Reviewer 1 Report

Holstein and colleagues identified potential post-translational modifications (PTMs) predicted based on mutations affecting potentially protein domains of site where these PTMs could occur. They investigated thousands of tumor sample sequences from the TCGA cohort databank. The novelty is to focus on PTM diruptive mutations in matrisome genes. Besides of course the hydroxyproline PTM mostly found in collagens (=matrisome), other PTMs were enriched in matrisome over genomic background. Highlighting a potential importance of these PTM disrupting extracellular matrix function and potentially tumorgenisis or even ultimatively metastasis. 

The study is well designed, performed, and documented. 

I only have one concern. Comparing matrisome to genome is good. But how about another "negative control". Could the authors randomly chose the same number of genes as the matrisome? And sample like this 10 times to see whether the matrisome subset performs better or worse as the randomly picked genes. Similarly, would it be possible to have like a "positive control".. like the known genes with PTMs in cancer, such as p53 etc. That would be interesting to see how well the matrisome performs compared to these "gold standards" chosen genes. 

Author Response

We thank the reviewer for the positive and constructive comments and for the possibility to further improve our manuscript.

As for the concerns raised:

1) we actually had - during the analyses - already tested what the reviewer suggested, that is, checking PTMmut distributions against random picks of the rest of the genome, each set being equal to the matrisome in size. We did not report, nor mention, this part as we saw it as of no additional value to the findings presented (and, hence, we removed it from the public code). But we are happy to put it back into the manuscript and to briefly discuss it. As evident, the assumption that the matrisome accumulates more PTMmut of specific types  holds even against random samples of the genome, repeated 100 times with replacement (see "Reviewer Table 1"). We have now added a mention to this in the manuscript (as "data not shown"), page 5, line 4-6.

2) This is an interesting point which we addressed at the specific PTMmut position level when dealing with prospective "hotspots"(page 5, lines 6-11 and  page 9, the paragraph after Figure 3). However, we didn't consider comparing the matrisome as a whole against "golden standards" of PTMmut to get a sense of the quantitative effect on the matrisome at a more general level. To this aim we have now constructed an "average matrisome metagene" (mean number of matrisome PTMmut across PTM types) and compared it against the most frequently affected genes identified by, e.g., Narayan et al. (https://www.ncbi.nlm.nih.gov/pmc/articles/PMC4864925/). As evident from "Reviewer Figure 1" (and, also, as expected), the matrisome trails behind the most abundantly mutated genes - all being also famous cancer drivers such as TP53, PTEN, MYC, NPM1, etc. but, as a whole across PTMs, it reaches the same levels of overall PTMmut as AKT1. In our opinion, this consolidates the observation that the overall contribution of the matrisome to the pool of cancer PTMmut is low (at best as much as the lowest of the cancer drivers assessed), in line with the observations we made on the probable negative selection operating on these mutations. Since we are unsure as to the contribution of this piece of information to the results already presented in the article, we have decided not to append it to the revised version.

Reviewer 2 Report

The study by Holstein et al., analyze mutations affecting post-translational modification (PTM) sites of matrisomal proteins in human tumor samples. In general consequences of mutations in genes coding for extracellular proteins remain underdressed, and in particular little work has been done on PTM-disruptive mutations of these in cancer. The authors here took a bioinformatic approach to investigate this. The study is well presented and should spark an interest in functionally investigating the consequences of mutations disrupting PTMs of matrisomal proteins in cancer.

I have a few comments and questions.

Do some of the mutations affecting PTMs of the matrisome also occur in healthy individuals or are all mutations truly unique for cancer?

For the PTM-mutation counts per cancer type, this should be normalized to the total number of mutations, e.g. SKCM has high mutational burden and only based on that it would be expected that the total number of PTM-mutations in genes encoding matrisomal proteins would be higher than for tumors with lower mutational burden.

Figure 3, the scaling of circles indicating number of occurrences should be adjusted. 

Do the sequencing analyzed derive from whole tumor samples e.g. both cancer cells and non-transformed cells present in the tumors? If this is the case, this should be clearly stated. 

Care should also be taken to not conclude too much based on sequencing analysis. It is not clear if some the genes studied are expressed to any significant level in some of the tumors.

There is a speculation that some of the PTM mutations may lead to lower fitness of clones carrying these mutations. I agree that this certainly appears to be the case based on the analyses and calculations presented. However, I miss the aspect of intracellular consequences of the mutations. It is possible that disrupting some PTMs could lead to impaired secretion and cellular stress. 

The last part of the results on functional characterization of matrisome PTMmut is without experimental testing a bit unsubstantiated and although interesting in some aspects, I think that this part should be shortened and moved to the discussion. 

Author Response

We thank the reviewer for the positive and constructive comments and for the possibility to further improve our manuscript.

As for the concerns raised:

1) The PTMmut reported are, per design, only calculated from cancer (4485 entries from metastatic samples and 38248 primary samples - note that entries might be duplicates since these numbers have not been worked through the checks we ran for the actual data in the manuscript). Indeed, the "mc3.v0.2.8.PUBLIC.xena" (available at https://xenabrowser.net/datapages/?dataset=mc3.v0.2.8.PUBLIC.xena&host=https%3A%2F%2Fpancanatlas.xenahubs.net&removeHub=https%3A%2F%2Fxena.treehouse.gi.ucsc.edu%3A443) and derived from Ellrott K et al. (PMID 29596782) includes data from tumor samples only. The question is, however, very relevant and we have decided to check the PTMmut reported across TCGA samples in the NCI-GDC Data Portal (https://portal.gdc.cancer.gov/), focusing exclusively on samples tagged as healthy derived from solid tissues, blood or bone marrow. We found that 143 PTMmut entries out of total 1509 entries (9.5%) are not found in any healthy samples, with additional 215 (14.2%) more frequent in tumors than in normal samples, for a total of approx. 24% tumor-biased PTMmut entries. It is, however, extremely difficult to assess the real prevalence of these mutations into the "healthy" population, as the size of these samples is much below the tumor samples AND below the tumor/healthy ration in the normal population, making this comparison intriguing but poorly informative. On these basis, we have decided not to show these data in toto in the manuscript but we have included a statement on page 12, line 21-23, and added a new supplementary table (Table S10).

2) We agree with this point and we have now added a relative frequency per cancer (% of total per cancer) tab to Table S2.

3) We agree that scaling of the circles could be adjusted to remove overlaps. On the other hand, given the predominance of single-case mutations among PTMmut, we believe that the majority of them would disappear. Since Figure 3 refers to data provided already in Table S2, we would be against reformatting this figure.

4) TCGA samples are bulk, hence it is likely that some mutations might be coming from healthy surrounding cells (e.g., stromal and immune cells). To our knowledge, it is currently impossible to define the origin of any of these mutations (unlike what possible with deconvolution of RNAseq data) but we had accounted for this in Page 11 already, as well as in our previous manuscript (Ref. 40 in the revised version).

5) We agree with the reviewer that the effect of any PTMmut would better be investigated by also taking into consideration the expression of the gene affected by mutation. We surmise, however, that is nearly impossible to define an "absolute" expression threshold on which basis to define whether an affected gene is "present" or not in the pool of the genes characterizing the given tumor. In an attempt to approach this problem, however, we have confronted PTMmut genes with matrisome "landmark" genes we previously identified (https://www.mdpi.com/1422-0067/21/22/8837) as bona fide markers of a given cancer type. Results show that 9 genes affected by PTMmut are also landmark genes, adding to the evidence of a more functional role for these mutations in the cancer matrisome that we had already laid out, e.g., in Page 12 lines 12-14. We have now added a brief discussion on this aspect in Page 12 lines 22-24.

6) We thank the reviewer for raising this point. In fact, it is possible that ECM proteins harboring PTMmut would undergo defective processing (e.g., in the Golgi apparatus) and be accumulated rather than secreted, leading to cell stress and death. This, in turn, would reduce the fitness of the clone harboring the mutation and thus concur to the same phenomenon of lower fitness as we observed in the data. We deem this even more likely for PTMmut affecting, in example, ubiquitylation, as this PTM has a major role in protein processing. We have now acknowledged this possibility in Page 9, line 40.

7) We agree with the Reviewer that this part would be better substantiated by wet lab biological data and, as it is now, it is quite speculative. We do believe, however, that the numbers we have gathered about what we discuss in this section are robust and serve the purpose to stimulate discussions and further studies aiming at evaluating these mutations in vitro/in vivo and, as such, we would prefer leaving it as it is.